# Effector memory CD8 T cell response elicits Hepatitis E Virus genotype 3 pathogenesis in the elderly

**Hicham El Costa**[1,2]*, **Jordi Gouilly**[1], **Florence Abravanel**[1,2], **Elmostafa Bahraoui**[1],
**Jean-Marie Peron**[3], **Nassim Kamar**[1], **Nabila Jabrane-Ferrat**[1‡], **Jacques Izopet**[1,2‡]

1 Infinity—Université Toulouse, CNRS, Inserm, Toulouse, France, 2 Laboratoire de Virologie, Centre National de référence HEV, Institut Fédératif de Biologie, CHU Toulouse, Toulouse, France, 3 Département de Gastroentérologie, CHU Toulouse, Toulouse, France

‡ These authors are joint senior authors on this work.
* hicham.el-costa@inserm.fr

**Data Availability Statement:** All relevant data are within the manuscript and its Supporting Information files.

## Abstract

Genotype 3 Hepatitis E virus (HEV-3) is an emerging threat for aging population. More than one third of older infected patients develops clinical symptoms with severe liver damage, while others remain asymptomatic. The origin of this discrepancy is still elusive although HEV-3 pathogenesis appears to be immune-mediated. Therefore, we investigated the role of CD8 T cells in the outcome of the infection in immunocompetent elderly subjects. We enrolled twenty two HEV-3-infected patients displaying similar viral determinants and fifteen healthy donors. Among the infected group, sixteen patients experienced clinical symptoms related to liver disease while six remained asymptomatic. Here we report that symptomatic infection is characterized by an expansion of highly activated effector memory CD8 T (EM) cells, regardless of antigen specificity. This robust activation is associated with key features of early T cell exhaustion including a loss in polyfunctional type-1 cytokine production and partial commitment to type-2 cells. In addition, we show that bystander activation of EM cells seems to be dependent on the inflammatory cytokines IL-15 and IL-18, and is supported by an upregulation of the activating receptor NKG2D and an exuberant expression of T-Bet and T-Bet-regulated genes including granzyme B and CXCR3. We also show that the inflammatory chemokines CXCL9-10 are increased in symptomatic patients thereby fostering the recruitment of highly cytotoxic EM cells into the liver in a CXCR3-dependent manner. Finally, we find that the EM-biased immune response returns to homeostasis following viral clearance and disease resolution, further linking the EM cells response to viral burden. Conversely, asymptomatic patients are endowed with low-to-moderate EM cell response. In summary, our findings define immune correlates that contribute to HEV-3 pathogenesis and emphasize the central role of EM cells in governing the outcome of the infection.

**Funding:** This work was supported by the Agence Nationale de Recherche sur le SIDA et les hépatites virales (contract number: ECTZ101971). The funder had no role in study design, data collection and analysis, decision to publish, or preparation of the manuscript.

**Competing interests:** The authors have declared that no competing interests exist.

## Author summary

The outcome of Genotype 3 Hepatitis E virus (HEV-3) infection differs among the elderly. Some patients develop severe forms of Hepatitis E while others remain asymptomatic. Nonetheless, parameters which can lead to severe *versus* silent infection are largely unknown. Therefore, we investigated immunological features of CD8 T cells in infected patients (aged ≥55) with similar viral determinants but distinct clinical outcomes. We show that drastic phenotypic changes were specifically observed within the effector memory (EM) compartment. Compared to asymptomatic patients, symptomatic ones display a strong activation of both HEV-3-specific and -nonspecific EM CD8 T cells associated with qualitative and quantitative alterations in cytokine production. In addition, EM cells are endowed with high cytotoxic capacity and have the ability to rapidly migrate to the liver. Finally, we report that the inflammatory response to HEV-3 infection shape EM cell activation and function in symptomatic elderly patients. In summary, our results present the first report demonstrating that the nature and the magnitude of EM CD8 T cell response play an important role in the outcome of HEV-3 infection in the elderly.

## Introduction

Hepatitis E Virus (HEV) is a positive-sense single-stranded RNA virus that belongs to the *Hepeviridae* family. Human cases of Hepatitis E are caused by *Orthohepevirus A* species which comprise 8 genotypes [1]. The Genotype 3 (HEV-3), widely prevalent in high-income countries, has a zoonotic transmission. In healthy young adults, HEV-3 infection follows mostly a clinically silent course and resolves spontaneously. However, the infection is often associated with adverse outcomes among the elderly. One third of these infected patients develop clinical symptoms ranging from acute icteric hepatitis to severe liver damage [2–4].

Host immune response, rather than the virus itself, seems to be involved in the burden of HEV-3 [5]. This issue has been highlighted in immunocompromised patients where both persistent infection and liver disease were associated with weak specific T cell response and increased expression of the programmed cell death (PD)-1 and cytotoxic T lymphocyte antigen (CTLA)-4 inhibitory receptors (iRs) [6]. Among T cell populations, CD8 T cells play a critical role in the immune response against HEV-3 infection. They massively infiltrate the liver during the acute phase of infection [7] and display a strong effector response characterized by the secretion of antiviral cytokines against well-conserved regions of the HEV-3 open reading frame 2 (ORF2) protein [8].

CD8 T cell population can be subdivided into four subsets based on the expression of the CD45RA leukocyte common antigen isoform and the chemokine receptor CCR7 [9]. Accordingly, the CD45RA$^+$CCR7$^+$ are naïve (N), the CD45RA$^+$CCR7$^-$ are effectors (E), the CD45RA$^-$CCR7$^+$ are central memory (CM) and the CD45RA$^-$CCR7$^-$ are effector memory (EM) cells. In addition to the well-characterized cytotoxic CD8 T subpopulation (Tc1), producing interferon (IFN)-γ and tumor necrosis factor (TNF)-α, growing evidence indicates the presence of an "alternative" CD8 T-cell subset (Tc2). Whether under physiological or pathological settings, Tc2 subset acquires the expression of type-2 cytokines such as interleukin (IL)-4 which influences the immune response [10].

The landscape of the CD8 T cell compartment is highly shaped with aging, placing elderly people at high risk of contracting infectious diseases [3,11–13]. Nonetheless, why some patients develop severe forms of Hepatitis E while others remain asymptomatic is still elusive. More specifically, parameters which can lead to severe *versus* silent infection, such as the

specificity and magnitude of the T cell response, are largely unknown. We therefore investigated the role of CD8 T compartment in elderly HEV-3-infected patients. Analyses of CD8 T cell subsets in patients with similar viral determinants revealed that symptomatic patients display a strong activation of both HEV-3-specific and -nonspecific EM cells with high cytotoxic profile. The activation of these cells was associated with elevated expression of the transcription factor T-Bet and contrasted with qualitative and quantitative defects in type-1 and type-2 cytokine production in EM cells. Our data also suggest that the inflammatory response to HEV-3 promotes bystander activation of EM cells, and increases their innate-like cytotoxicity and ability to migrate into the liver in CXCR3-dependent manner. Overall, our findings substantiate that the EM compartment contribute to HEV-3 pathogenesis in the elderly.

## Results

### Characteristics of study subjects

To investigate the role of CD8 T cells in HEV-3 pathogenesis, we enrolled thirty seven immunocompetent subjects (aged $\geq$ 55 years) including twenty two HEV-3-infected patients and fifteen healthy donors. Among HEV-3-infected patients, sixteen experienced clinical symptoms related to liver disease while six were asymptomatic. Both symptomatic and asymptomatic patients displayed similar HEV-3 load and subtype distribution (Table 1).

**Table 1. Patient characteristics.**

| | HEV-infected patients | | Healthy controls |
|---|---|---|---|
| | Symptomatic | Asymptomatic | |
| | n = 16 | n = 6 | n = 15 |
| Gender (m/f) | 11/5 | 1/5 | 9/6 |
| Age (y) | 65.3 ± 2.1 | 61.3 ± 1.6 | 61.9 ± 1.2 |
| Physical symptoms (%) | | | |
| Fever | 50 | 16.6 | 0 |
| Mucosal icterus | 100 | 0 | 0 |
| Colored urine | 100 | 0 | 0 |
| Discolored stools | 100 | 0 | 0 |
| Biochemical analysis | | | |
| ALT Level (IU/L) | 1863 ± 400.4 | 489.8 ± 219.7 | |
| AST Level (IU/L) | 1153 ± 295.6 | 111.2 ± 47.5 | |
| Bilirubin (μM) | 130.2 ± 24.3 | 10.3 ± 1.2 | |
| PR <50% | 4/16 | 0/6 | |
| Anti-HEV IgM | | | |
| Positive (n) | 16 | 6 | 0 |
| Negative (n) | 0 | 0 | 15 |
| Anti-HEV IgG | | | |
| Positive (n) | 16 | 6 | 0 |
| Negative (n) | 0 | 0 | 15 |
| HEV RNA in plasma | 5.3 ± 0.3 | 4.1 ± 0.7 | |
| (log copies/ml) | | | |
| HEV-3 subtypes | 14/2/0/0 | 4/0/1/1 | |
| (3f/3c/3h/UD) | | | |

n, number of subject; m, male; f, female; y, years; ALT, alanine aminotransferase; AST, aspartate aminotransferase; PR, Prothrombin Ratio; UD, undetermined.

Data represent mean values ± S.E.M.

## Excessive activation of the EM compartment in symptomatic HEV-3 infection in the elderly

To assess whether HEV-3 infection shapes the phenotype of CD8 T cells, we analyzed the expression of subset markers (CCR7 and CD45RA), activation markers (HLA-DR and CD38), costimulatory receptors (CD27 and inducible T-cell costimulator (ICOS)), iRs (PD-1, T cell immunoglobulin mucin domain (TIM)-3 and Lymphocyte activation gene (LAG)-3) as well as memory markers (CD161 and CD127) on PBMC from all subjects. No significant differences were found in the proportion of CD8 T cells (Fig 1A). However, the frequency of the EM subset was significantly increased in symptomatic patients (Fig 1B and 1C). Moreover, drastic phenotypic changes were specifically observed within the EM compartment (Figs 1D and S1A). Compared to asymptomatic and healthy subjects, the EM compartment from symptomatic patients exhibited significantly higher proportions of HLA-DR, CD38, PD-1, TIM-3, LAG-3, CD27 and ICOS positive cells (Fig 1D). Conversely, there was a lower proportion of CD127 and CD161 positive cells (Fig 1D). The density expression (ΔMFI) of HLA-DR, CD38 and PD-1 was also higher in EM cells from symptomatic patients (S1B Fig). On the other hand, EM cells from asymptomatic patients displayed less pronounced changes and significant differences were found only for CD38, PD-1 and CD127 when compared to healthy subjects (Fig 1D). The phenotypic signature of symptomatic patients is reminiscent of exacerbated activation [14–16]. In line with this notion, expression levels of HLA-DR and CD38 were negatively correlated to those of CD127 and CD161, and positively correlated to those of PD-1, ICOS and CD27 (S1C Fig). Furthermore, the coexpression of PD-1/TIM-3/LAG-3 was restricted to EM cells from symptomatic patients (S1D Fig). To further corroborate this robust activation, we measured the expression of the proliferation antigen Ki-67 and anti-apoptotic protein Bcl-2 on EM cells [17]. Symptomatic patients showed sharp increase of Ki-67 expression and downregulation of Bcl-2 (Figs 1E and S1E). By contrast, asymptomatic patients exhibited a resting Ki-67$^{low}$Bcl-2$^{high}$ phenotype (Fig 1E). Finally, to visualize the global pattern of EM subsets during the acute phase of the infection, we included all the aforementioned markers in a principal component analysis (Fig 1F). Infected patients clustered into two distinct phenotypes with an increased/exacerbated immune activation for symptomatic patients and a low activation threshold for asymptomatic ones. Overall, our results clearly demonstrate that symptomatic patients display a massive expansion of activated EM cells. They also suggest that the activation threshold of EM subsets is linked with the outcome of HEV-3 infection in the elderly.

## Coordinated expression of HLA-DR, CD38 and PD-1 discriminates symptomatic HEV-3 infection in the elderly

HLA-DR, CD38 and PD-1 expression displayed the highest impact on the principal component 1 (PC1) when compared to the other principal component analysis loadings (Fig 1F). Since phenotypic changes in the EM compartment are correlated with the activation status, we focused our analysis on HLA-DR, CD38 and PD-1 as representative markers to provide a concise immune signature discriminating symptomatic from asymptomatic patients. EM compartment from symptomatic patients was strikingly marked by an increased frequency of cells coexpressing HLA-DR/CD38/PD-1 (Fig 2A, 2B and 2C). This feature was associated with a significant decrease in the frequency of cells expressing none of these markers (Fig 2B and 2D). Conversely, the EM subset from asymptomatic patients was characterized by a slight increase in cells expressing a single activation marker further highlighting their mild activation status (Fig 2B and 2E). Collectively, our results show that HLA-DR/CD38/PD-1 coexpression is a relevant signature to probe the activation status of EM cells during acute HEV-3 infection and predict the disease outcome in the elderly.

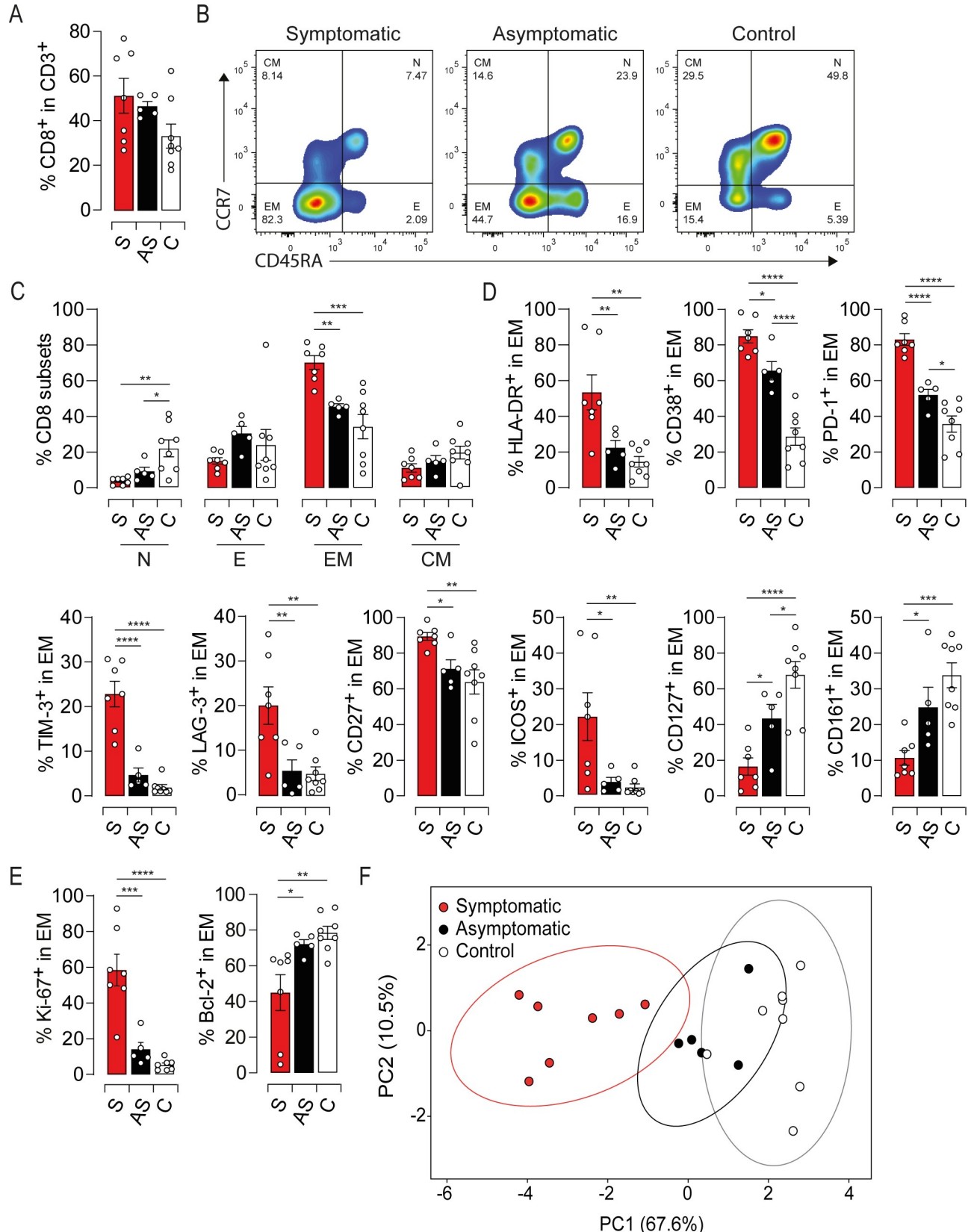

**Fig 1. Phenotypic landscape of EM cells during symptomatic HEV-3 infection.** (A) Proportion of CD8 T cells in CD3+ cells. (B) Representative dot plots and (C) mean percentage of CD8 T subsets: naïve (N), effector (E), effector memory (EM) and central memory (CM). (D) Expression of surface and (E) intracellular markers in EM cells. (F) Principal Component Analysis of EM cell phenotype based on HLA-DR, CD38, PD-1, TIM-3, LAG-3, CD27, ICOS, CD127 and CD161 expression. The probability that a new observation within the same group fall in the ellipse is 0.95. Data represent mean values±S.E.M. *P<0.05; **P<0.01; ***P<0.001; ****P<0.0001. Symptomatic patients (S, red), asymptomatic patients (AS, black) and healthy controls (C, white).

## Loss of polyfunctional type-1 cytokine production in highly activated EM cells from symptomatic elderly patients

To investigate whether the excessive activation is associated with functional alterations [15,18], we first assessed the ability of EM cells to produce type-1 cytokines. PBMC were subjected to a brief polyclonal stimulation and the production/coproduction of TNF-α, IFN-γ and IL-2 was assessed by intracellular staining. Symptomatic patients displayed a drastic drop in the global TNF-α production that was not corrected under prolonged stimulation suggesting an intrinsic defect in EM cells (Figs 3A, S2A and S2B). By contrast, the overall amount of IFN-γ was increased while IL-2 was decreased respectively but did not reach significant difference between study groups (S2C and S2D Fig). Furthermore, symptomatic patients displayed a loss of polyfunctionality highlighted by a decrease in IFN-γ/TNF-α double positive population and an increase in the frequency of monofunctional cells producing only IFN-γ (Fig 3B, 3C and 3D). We next investigated the link between this functional bias and the activation status of the EM compartment using t-SNE analysis of HLA-DR, CD38, PD-1, IFN-γ, TNF-α and IL-2 expression. Of note, the expression of activation/exhaustion markers was not affected by the polyclonal stimulation (S2E Fig). t-SNE plots clustered EM cells into two major groups: symptomatic patients and, asymptomatic or healthy donors (Fig 3E, upper panel). Moreover, EM cells expressing high levels of HLA-DR, CD38 and PD-1 were dominated by monofunctional IFN-γ-producing cells in symptomatic patients (Fig 3E, lower panel). To validate these observations, we characterized the expression of HLA-DR, CD38 and PD-1 in cells which fail to coproduce IFN-γ/TNF-α in symptomatic patients. More than 70% of IFN-γ-monofunctional cells coexpressed these markers (Fig 3F). We also assessed the coproduction of type-1 cytokines based on the expression/coexpression of the activation markers. Functional impairments were observed only within the major subset of EM cells coexpressing HLA-DR/CD38/PD-1 (S2F, S2G and S2H Fig). Thus, highly activated EM cells exhibit a polyfunctionality loss while slightly activated cells are fully functional. To further link the immune status of EM cells to viral burden, we assessed the activation threshold and cytokine production in three symptomatic patients at the convalescence period. One month after the onset of symptoms, HLA-DR/CD38/PD-1 expression drastically decreased compared to the early stage of the infection (S2I Fig). In addition, TNF-α production was restored to healthy control levels. Interestingly, the recovery of EM cell response coincided not only with viral clearance but also with a decline in the liver enzyme levels and resolution of symptoms (S2I Fig), strengthening the notion that EM cells are responding to HEV-3 infection. Taken together, our results show that the functional impairment of the EM compartment combined with excessive activation underlie the acute phase of symptomatic HEV-3 infection.

## Partial commitment of EM compartment to type-2 cells in symptomatic elderly patients

Symptomatic patients were characterized by an accumulation of an EM subset expressing low levels of CD8 (CD8L) (Fig 4A), alluding to Tc2 cells [10]. Therefore, we first monitored IL-4 production in EM compartment. IL-4 was mainly produced by the CD8L subset in

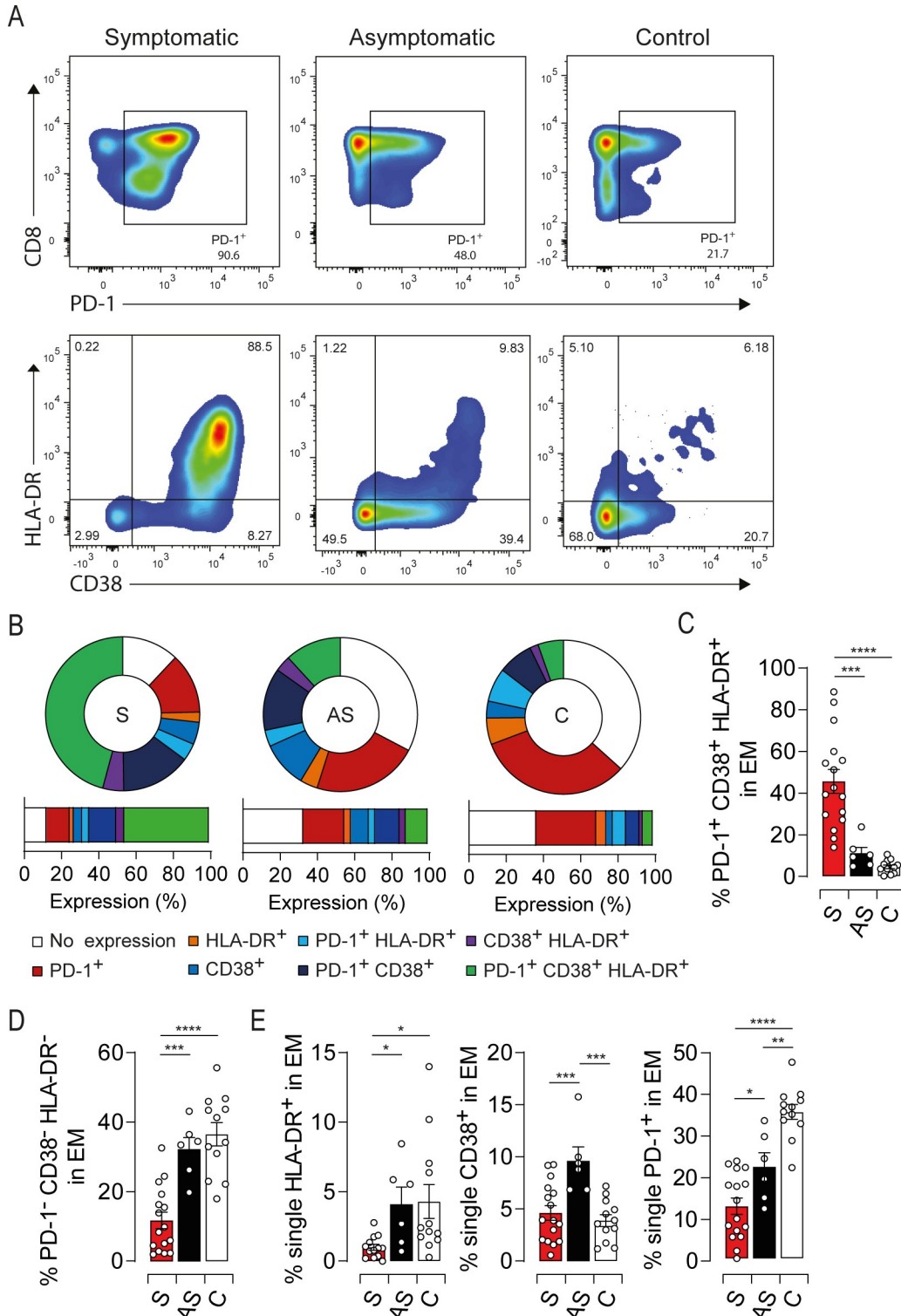

**Fig 2. HEV-3 symptomatic patients harbor sizeable proportion of highly activated EM cells coexpressing HLA-DR, CD38 and PD-1.** (A) Representative dot plots for HLA-DR, CD38 and PD-1 coexpression in EM cells. EM cells were segregated based on PD-1 expression (upper panel). CD38 and HLA-DR expression were illustrated in PD-1+ cells (lower panel). (B) Donut charts representing the mean percentage of cells identified by single/double/triple expression or absence (no expression) of PD-1, HLA-DR and CD38. (C, D and E) Bar graphs representing statistical analyses of the donut charts. Data represent mean values±S.E.M. $^{*}P<0.05$; $^{**}P<0.01$; $^{***}P<0.001$; $^{****}P<0.0001$ Symptomatic patients (S, red), asymptomatic patients (AS, black) and healthy controls (C, white).

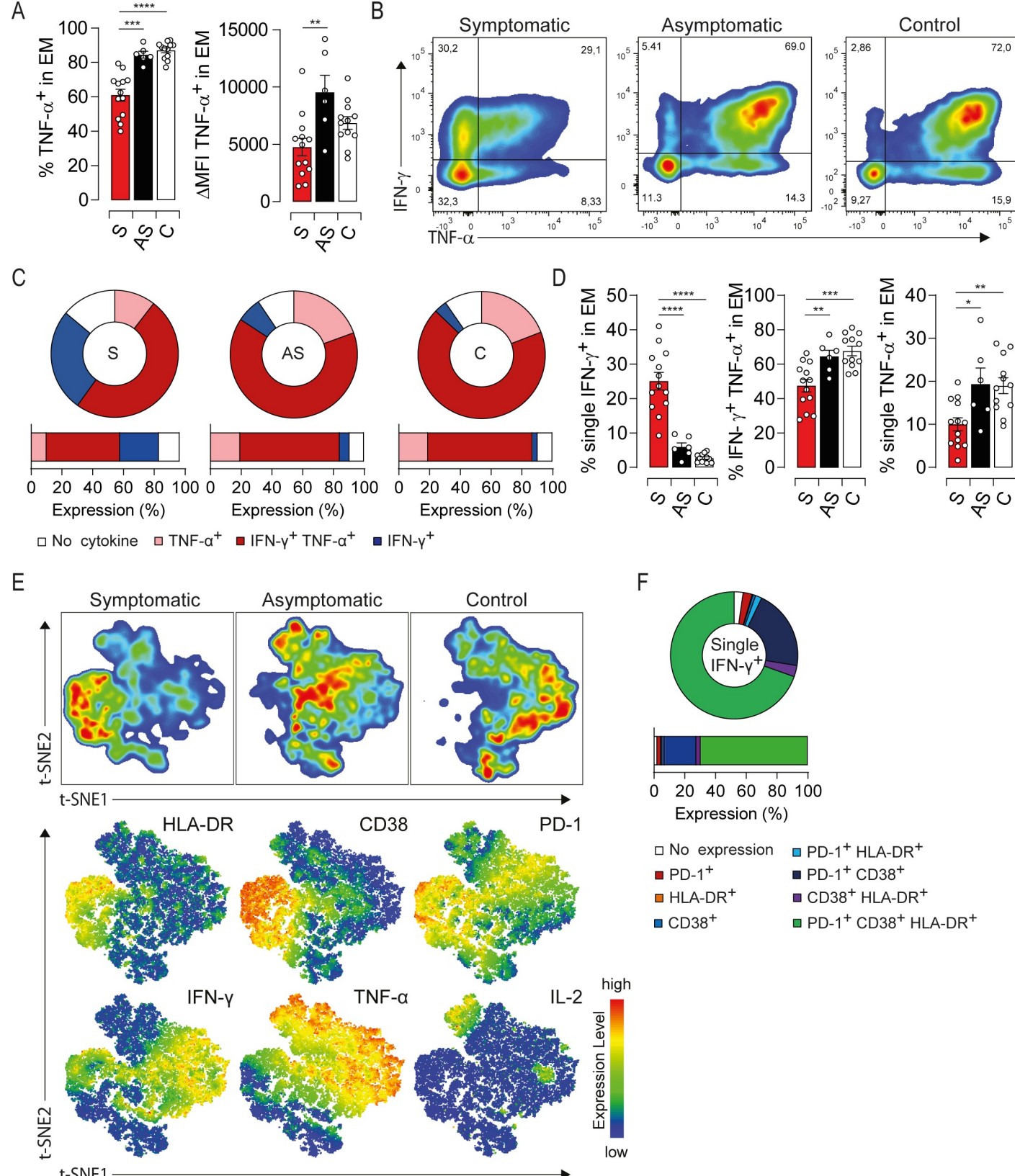

**Fig 3. Impairment of type-1 cytokine profile is associated with high activation status of EM cells in HEV-3 symptomatic patients.** (A) Left panel illustrates the frequency of EM cells producing TNF-α. Right panel shows ΔMFI of TNF-α in EM cells. (B) Representative dot plot for IFN-γ/TNF-α production in EM cells. (C) Donut charts representing the mean percentage of EM cells identified by single/double expression or absence (no cytokine) of IFN-γ and TNF-α. (D) Bar graphs representing statistical analyses of the donut charts. (E) t-SNE analysis of indicated markers in EM cells using three representative donors from each study groups. Upper panel shows the clustering of study groups based on the global expression profile of indicated markers. Lower panel shows expression profile of each individual marker. (F) Donut chart representing the mean percentage of cells expressing different combination of HLA-DR, CD38 and PD-1 in EM monofunctional IFN-γ-producing cells. Data represent mean values±S.E.M. *P<0.05; **P<0.01; ***P<0.001; ****P<0.0001. Symptomatic patients (S, red), asymptomatic patients (AS, black) and healthy controls (C, white).

symptomatic patients with nearly 60% of IL-4-producing cells, while only a moderate proportion was seen in asymptomatic ones (Figs 4B, 4C, S3A and S3B). Of note, polyclonal stimulation did not alter significantly IL-4 production in HEV-3 patients, further strengthening the notion that IL-4 is induced *in vivo* under pathological conditions (S3C Fig). Albeit being unusual in healthy subjects, similar spontaneous cytokine production was previously described during acute illness [19]. By contrast, IFN-γ production was decreased in CD8L subset compared to their counterpart expressing high level of CD8 (CD8H) (Fig 4D). Moreover, IFN-γ was barely detected in IL-4-producing cells indicating their commitment to the type-2 lineage (S3D Fig). Likewise, CD8L cells from symptomatic patients harbored significantly lower proportion of TNF-α-producing cells compared to asymptomatic or healthy donors (Fig 4E). In addition, IL-4 negatively correlated with TNF-α production (S3E Fig) implying its contribution to the impairment of type-1 cytokine production in symptomatic patients. We next assessed the relationship between IL-4 production and the activation status of CD8L cells. t-SNE analysis confirmed that the majority of CD8L cells belongs to the symptomatic group (Fig 4F). Furthermore, these cells exhibited high levels of HLA-DR, CD38 and PD-1 along with increased IL-4 and decreased IFN-γ productions (Fig 4E, lower panel). Indeed, more than 50% of IL-4-producing cells were HLA-DR/CD38/PD-1 triple positives in symptomatic patients, whilst cells expressing either one or none of these markers failed to produce IL-4 (Fig 4H). When extending our analysis to the convalescence period, the frequency and activation threshold of CD8L subset as well as IL-4 production declined to levels similar to healthy controls (S3F Fig), suggesting a viral contribution to the commitment of EM compartment to type-2 cells. Altogether, our results emphasize a skewing of the EM compartment towards a highly activated Tc2 profile that may shape type-1 cytokine production in symptomatic patients.

## T-Bet expression supports EM cells dysfunction in symptomatic elderly patients

To investigate molecular pathways that might govern EM cell functions, we assessed the expression of T-Bet and Eomes, two T-Box transcription factors that orchestrate Tc1 response, and Gata-3, a transcription factor involved in Tc2 differentiation [20]. While Eomes and Gata-3 expressions were not affected (S4A Fig), T-Bet was significantly higher in symptomatic patients compared to asymptomatic and healthy subjects (Figs 5A and S4B). Distinct expression of T-Bet was also observed within the same individual based on CD8 expression levels (S4C Fig). Moreover, T-Bet was barely detected in IL-4-producing cells (S4D Fig), further confirming the polarization of CD8L subset toward a Tc2 profile. We next examined the relationship between T-Bet expression and the activation/dysfunction of EM cells using t-SNE analysis. T-Bet positive cells clustered with those expressing high levels of HLA-DR, CD38 and PD-1 from symptomatic patients (Fig 5B). In agreement with this observation, the T-Bet positive subset from symptomatic patients was strikingly marked by high frequency of HLA-DR/CD38/PD-1 triple positive cells, while the T-Bet negative subset was dominated by cells expressing none of these markers (Fig 5C). Interestingly, T-Bet expression in symptomatic

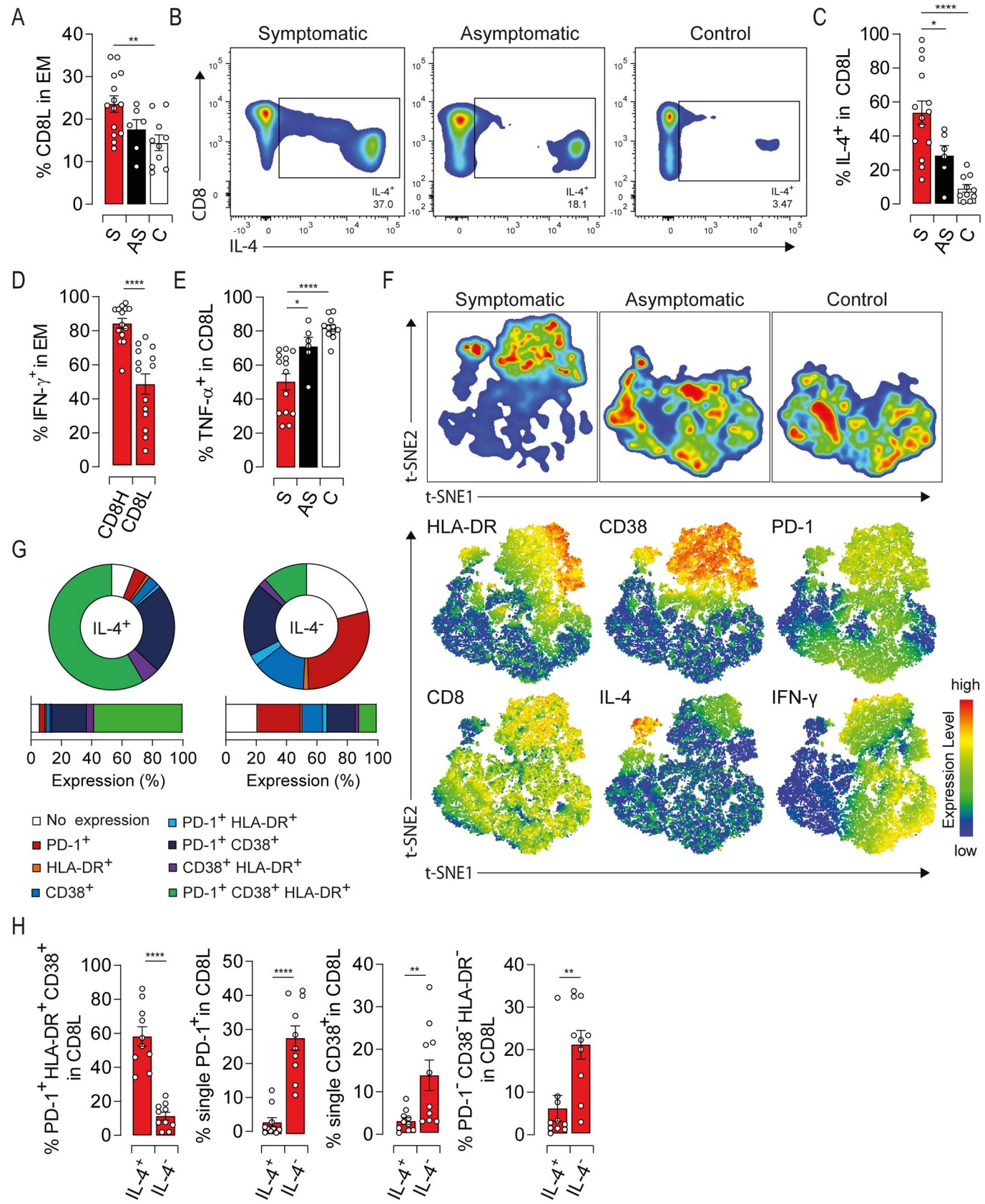

**Fig 4. Accumulation of a highly activated type-2-polarized EM CD8 subset producing IL-4 characterize symptomatic HEV-3 infection.** (A) Proportion of CD8 low subset (CD8L) within the EM compartment. (B) Representative dot plots of IL-4 staining in the EM subset. (C) Bar graphs representing the mean percentage of cells producing IL-4 in CD8L subset. (D) Bar graphs representing the mean percentage of cells producing IFN-γ in EM cells expressing high (CD8H) or low (CD8L) level of CD8. (E) Bar graphs representing the mean percentage of cells producing TNF-α in CD8L compartment. (F) t-SNE analysis of indicated markers in EM cells using three representative donors from each study groups. Upper panel shows the clustering of study groups based on the global expression profile of indicated markers. Lower panel shows expression profile of each individual marker. (G) Donut charts representing the mean percentage of cells identified by single/double/triple expression or absence (no expression) of HLA-DR, CD38 and PD-1 in CD8L cells producing IL-4 (IL-4$^+$) or not (IL-4$^-$) in symptomatic patients. (H) Bar graphs representing statistical analyses of donut charts. Data represent mean values±S.E.M. $^*$P<0.05; $^{**}$P<0.01; $^{***}$P<0.001, $^{****}$P<0.0001. Symptomatic patients (S, red), asymptomatic patients (AS, black) and healthy controls (C, white).

patients returned to resting levels after recovery, further linking T-Bet with the activation status of EM cells (S4E Fig). t-SNE plots also showed that T-Bet overexpression is associated with a decrease in TNF-α production (Fig 5B, lower panel). Indeed, T-Bet positive EM cells from symptomatic patients displayed high frequency of monofunctional IFN-γ-producing cells along with a decrease in TNF-α single positive population (Fig 5D). Asymptomatic and healthy subjects showed similar expression profiles but to a lesser extent than symptomatic patients (S4F, S4G, S4H and S4I Fig). To unravel the role of T-Bet in EM cells-mediated liver injury during HEV-3 infection, we assessed the expression of T-Bet-regulated factors namely CXCR3, a chemokine receptor involved in the recruitment of immune cells to the liver, granzyme B, a mediator of cell death and inflammation, and CD122, the IL-2 and IL-15 receptor β-subunit responsible for IL-15 responsiveness [20,21]. Compared to asymptomatic and healthy subjects, EM cells from symptomatic patients have higher surface expression of CXCR3 and CD122 and produce more granzyme B (Fig 5E). The increase in the level of these proteins was associated with both activation status of EM cells and T-Bet expression (Fig 5F and 5G). More importantly, the frequency of activated EM cells bearing CXCR3 was markedly high in symptomatic patients suggesting a higher ability to infiltrate the liver (Fig 5H). Collectively, our results indicate that T-Bet overexpression might play a pivotal role in HEV-3 pathogenesis by promoting the recruitment into the liver of highly activated EM cells endowed with increased cytotoxic capacity and altered cytokine production.

## EM cells activation in symptomatic elderly patients is not restricted to HEV-3-specific cells

Given that memory T cell response to viral infection can occur independently of cognate antigens [22], we assessed the antigenic specificity of activated EM cells in our study groups.

Growing evidence suggest that active response to HEV-3 is directed against peptide sequences within ORF2 (S5A Fig) [23–25]. We, therefore, identified HEV-3-specific EM cells (EM$_{HEV-3}$) producing IFN-γ following stimulation with a pool of ORF2-derived peptides. No IFN-γ-producing cells were detected in the control group, confirming the specificity of ORF2 stimulation (Fig 6A). As reported previously [8], the proportion of EM$_{HEV-3}$ cells ranged from 1 to 4% of the global EM compartment (Fig 6A and 6B). Compared to asymptomatic patients, EM$_{HEV-3}$ cells from symptomatic ones displayed lower production of TNF-α along with high expression of HLA-DR, CD38 and PD-1 (Figs 6C, 6D, S5B and S5C). These observations are in agreement with the global immune profile of EM cells from symptomatic patients. They point also to a bystander effect of HEV-3-unrelated CD8 T cells, given the frequency of EM$_{HEV-3}$ and the overall skewing of the EM compartment. To challenge this hypothesis, we monitored the activation status of EM cells specific to hCMV (EM$_{hCMV}$) and EBV (EM$_{EBV}$) using an HLA-A2 pentamers refolded with a CMV-pp65 or EBV-BMLF-1 epitopes. Both symptomatic patients and healthy controls displayed comparable frequencies of EM$_{hCMV}$ and EM$_{EBV}$ cells (Fig 6E). Interestingly, HEV-3-unrelated EM cells from symptomatic patients exhibited high activation status supported by increased expression of HLA-DR, CD38, PD-1

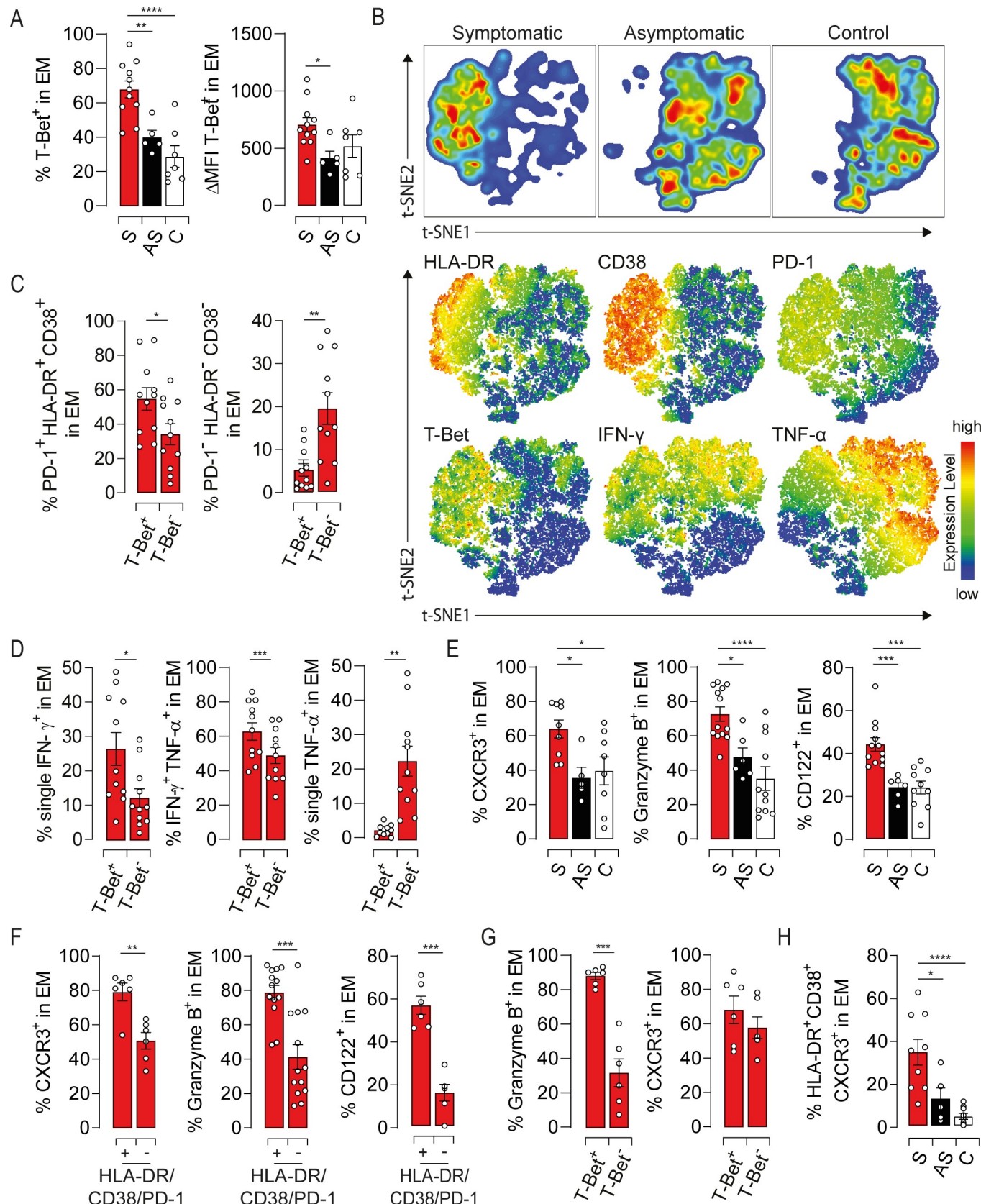

**Fig 5. T-bet overexpression governs the phenotypic and functional bias of EM cells in symptomatic HEV-3 infection.** (A) Left panel illustrates the mean percentage of EM cells expressing T-bet and right panel shows ΔMFI of T-Bet in EM cells. (B) t-SNE analysis of indicated markers in EM cells using three representative donors from each study groups. Upper panel shows the clustering of study groups based on the global expression profile of indicated markers. Lower panel shows expression profile of each individual marker. (C) Frequency of cells identified by triple expression or no expression of PD-1, HLA-DR and CD38 in the T-Bet positive (T-Bet$^+$) and negative (T-Bet$^-$) subsets in symptomatic patients. (D) Bar graphs illustrating the mean percentage of EM cells identified by single/double expression of IFN-γ and TNF-α in T-Bet$^+$ and T-Bet$^-$ subsets in symptomatic patients. (E) Mean percentage of cells expressing indicated markers in EM subsets. (F and G) Frequency of EM cells expressing indicated markers in (F) HLA-DR/CD38/PD-1 triple positive and triple negative subsets or (G) T-Bet$^+$ and T-Bet$^-$ subsets in symptomatic patients. (H) Frequency of cells identified by the triple expression of HLA-DR, CD38 and CXCR3 in the EM compartment. Data represent mean values±S.E.M. $^*$P<0.05; $^{**}$P<0.01; $^{***}$P<0.001; $^{****}$P<0.0001. Symptomatic patients (S, red), asymptomatic patients (AS, black) and healthy controls (C, white).

and Ki-67, compared to healthy controls (Figs 6F, S5D, S5E, S5F and S5G). TNF-α and IL-4 production followed also the global immune profile of EM cells although some EM$_{hCMV}$ and EM$_{EBV}$ failed to produce IL-4 (Fig 6F). Of note, viral DNA was undetectable in patients' sera, excluding a transient hCMV or EBV reactivation. Together, our findings imply that bystander cell activation contributes to the drastic alteration of the EM compartment in symptomatic patients.

## Inflammatory response to HEV-3 infection shapes EM cells activation and pathogenesis in symptomatic elderly patients

Most of liver disease are characterized by an inflammatory response including cytokines and chemokines secretion [21,26]. These latter are causal factors in nonspecific T cell activation and in orchestrating the sequential influx of immune cells into damaged tissue. To address this issue during HEV-3 infection, we sought to measure IFN-α2, IL-12p70, IL-15 and IL-18 in the plasma of infected patients. No difference was observed in IFN-α concentration (Fig 7A) while IL-12 was not detected. However, IL-15 and IL-18 were significantly increased in symptomatic patients compared to asymptomatic ones (Fig 7A). Of note, these cytokines were undetectable in the HEV-3-infected hepatocyte cell line culture supernatant. In addition to their role in bystander activation, signaling through IL-15 and IL-18 receptors was previously shown to modulate the expression of natural killer receptors (NKRs) on EM cells thereby triggering their innate-like cytotoxicity and affecting the outcome of the infection [22,27,28]. Therefore, we examined the expression of a wide range of activating receptors on EM cells. Except NKG2D, NKRs were barely detected in the EM compartment (Figs 7B and S6A). EM cells from infected patients displayed a slight increase in the frequency of NKG2D positive cells compared to healthy controls (Fig 7B). However, symptomatic patients harbored a considerable frequency of highly activated EM bearing NKG2D with a high cytotoxic potential (Fig 7C and 7D). Similar expression profiles of NKG2D and granzyme B were observed in HEV-3-specific and unrelated EM cells (Figs 7E and S6B). Owing to the role of NKG2D in enhancing the cytotoxic response of CD8 T cells in TCR-dependent and -independent manner [29], our data suggest that NKG2D might contribute to HEV-3 pathogenesis in IL-15-dependent manner. We next evaluated the plasmatic levels of chemokines in infected patients with a particular focus on CXCR3 ligands [21]. CXCL9, CXCL10 were significantly increased in symptomatic patients (Fig 7F), implying a massive attraction of EM cells into the liver through the CXCR3-mediated chemotaxis (Fig 5E, 5F, 5G and 5H). CCL4 might be also implicated although it was not detected in some patients (Fig 7F). Furthermore, we observed elevated levels of CXCL8 and CCL2 that promote the recruitment of non-adaptive immune cells to the liver such as monocytes, the main source of IL-15 and IL-18 during viral infections (Fig 7G) [30]. Infected hepatocytes may also drive EM and monocytes recruitment by the production of CXCL10 and CXCL8 (S6C Fig). Collectively, our data suggest that the inflammatory network

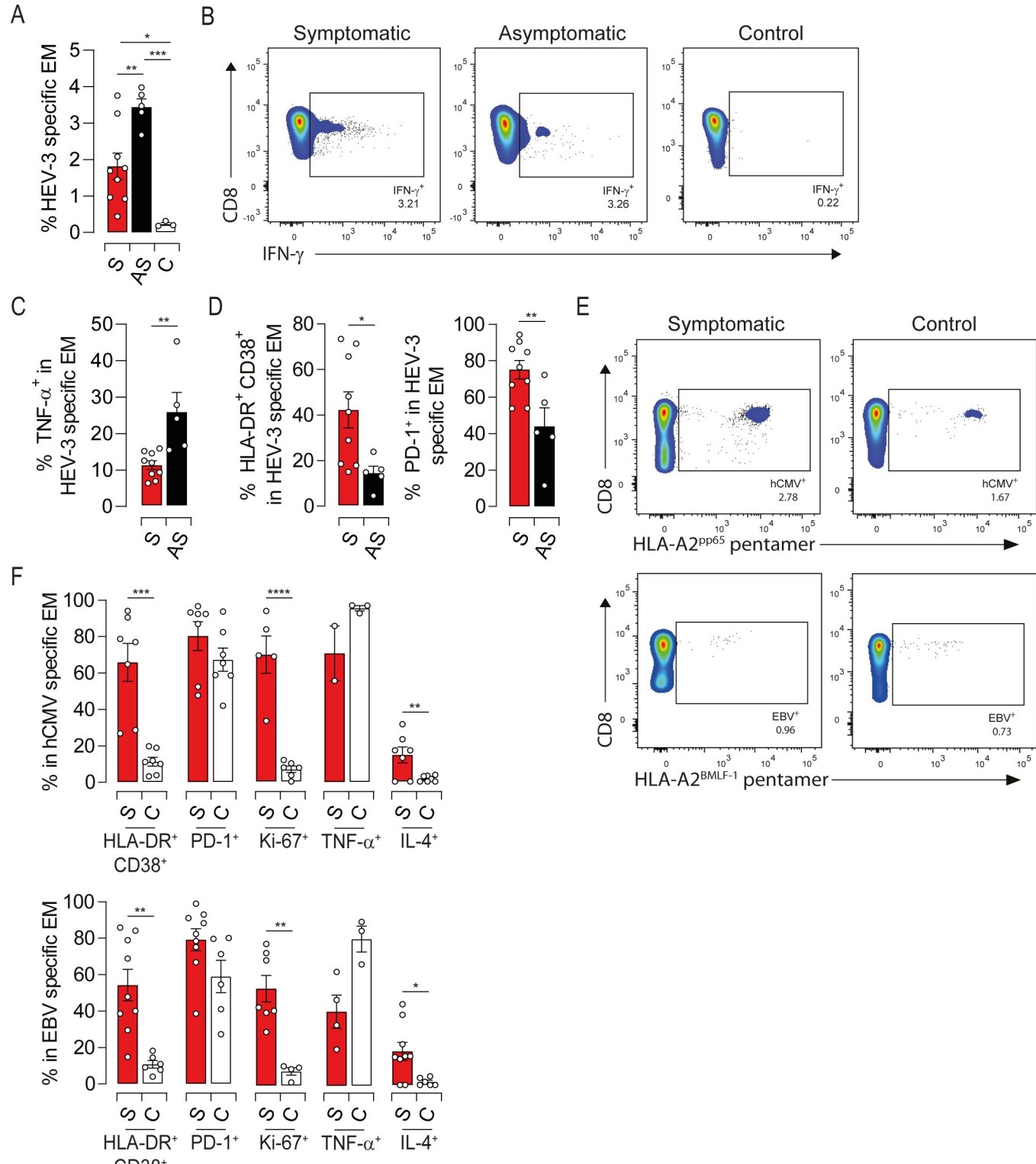

**Fig 6. Both virus-specific and -unrelated EM cells display an activated phenotype during symptomatic HEV-3 infection.** (A) Proportion of IFN-γ-producing cells (HEV-3-specific) upon stimulation with ORF2 peptides. (B) Representative dot plots of HEV-3-specific EM cells. (C) Proportion of TNF-α-producing cells in HEV-3-specific EM cells. (D) Mean percentage of cells identified by HLA-DR/CD38 coexpression (left panel) or PD-1 expression (right panel) within HEV-3-specific EM cells. (E) Representative dot plots of hCMV-specific (upper panel) and EBV-specific (lower panel) EM cells identified respectively by HLA-A2 pentamer-pp65[495-504] and HLA-A2 pentamer-BMLF-1[259−267]. (F) Mean percentage of cells expressing indicated markers in hCMV-

specific (upper panel) and EBV-specific (lower panel) EM cells. Data represent mean values±S.E.M. *P<0.05; **P<0.01; ***P<0.001; ****P<0.0001. Symptomatic patients (S, red), asymptomatic patients (AS, black) and healthy controls (C, white).

during HEV-3 infection is involved in the activation and recruitment of EM cells into the liver thereby fostering tissue injury through the innate-like cytotoxicity of these cells.

## Alterations within the EM compartment are the hallmark of symptomatic HEV-3 infection

Finally, to understand whether the pathogenesis of HEV-3 in the elderly is similar to that in younger patients, we investigated key features of EM cells in younger infected patients aged 27 to 47 years (Table 2). Compared to sex- and age-matched healthy control, no difference were found in the proportion of CD8 T cell subsets (Figs 8A and S7A). As in the context of aging, most changes were observed within the EM compartment (S7B Fig). Younger patients exhibited high activation status supported by an increase in HLA-DR, CD38, PD-1 and Ki-67 expression (Fig 8B) and contrasted with a loss in polyfunctional type-1 cytokine production (Fig 8C). IL-4 production was also increased although the frequency of CD8L subset remained unchanged (Fig 8D). By contrast, the expression of T-Bet and T-Bet-regulated factors did not differ from healthy controls (Fig 8E).

When compared to older patients, younger ones displayed lower frequency of EM cells and CD8L subset with less cells expressing granzyme B and IL-4 (Fig 8A, 8D and 8E). The levels of inflammatory chemokine were also diminished in young patients suggesting reduced recruitment of immune cells into the liver (Fig 8F). Although younger patients share with elderly ones some features of EM cell response, our data delineate differences that might explain why elderly patients are at high risk to develop severe form of Hepatitis E.

## Discussion

This study aims to determine mechanisms underlying HEV-3 pathogenesis in the elderly. We have achieved this by comparing key features of peripheral CD8 T cells from patients with distinct clinical outcomes. Among CD8 T cell subsets, the EM compartment of symptomatic patients exhibited deep phenotypic and functional alterations during the acute phase of the infection, which returned to homeostasis following disease resolution and viral clearance. Since HEV-3 *per se* is noncytopathic, our findings are consistent with the hypothesis that HEV-3 pathogenesis is triggered by EM cell responses.

The most striking pattern was the activation and the proliferation of HEV-3-related and -unrelated EM cells. Although immune response to hepatitis virus is initiated by antigen-specific T cells, global CD8 T cells activation can occur resulting in severe hepatitis [22,31–33]. Cross-reactive T-cell responses between heterologous viruses or against self-antigens has been suggested to be fairly extensive [34,35]. This issue is unlikely at play during HEV-3 infections since both $EM_{hCMV}$ and $EM_{EBV}$ cells were activated and displayed functional alterations. In addition, HEV-3 epitopes do not exhibit significant homology with amino acid sequence from these viruses. Pro-inflammatory cytokine microenvironment during acute infection, depletion/accumulation of some metabolites and/or innate-like functions of CD8 T cells can also trigger the activation of EM cells in the absence of cognate antigens as well as their functional skewing [23,26,30,33,36]. For instance, IL-15 and IL-18, alone or in combination with other factors, are able to induce rapid bystander activation of EM cells, underscored by the expression of HLA-DR, CD38 and Ki-67, regardless of the virus they target [22,30,37]. Accordingly, our data suggest that the pro-inflammatory cytokine IL-15 and IL-18 are key drivers in EM

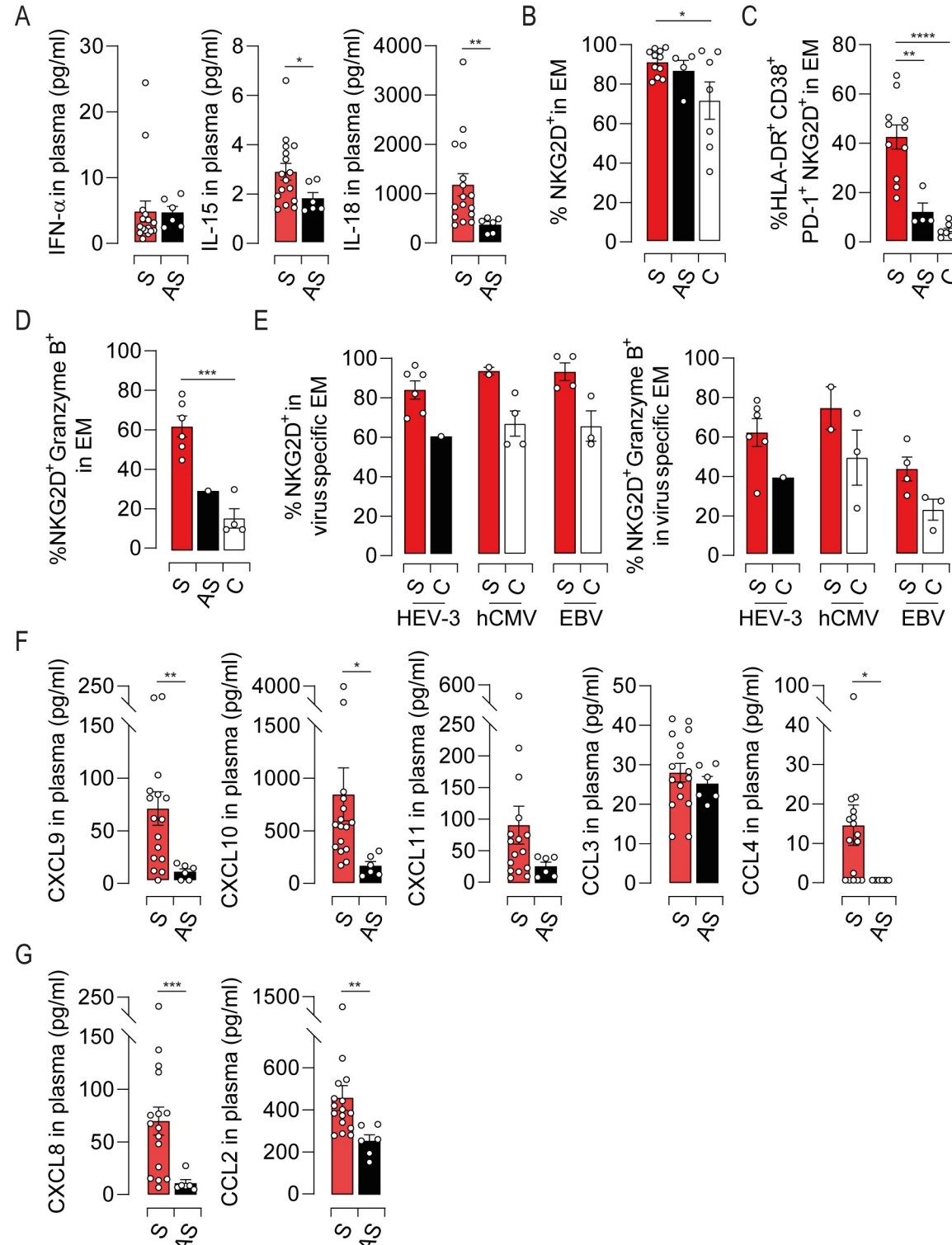

**Fig 7. Characterization of the inflammatory cytokine and chemokine landscapes during HEV-3 infection and its impact on the EM compartment.** (A) Plasmatic levels of indicated cytokines. (B) Mean percentage of cells identified by NKG2D expression in EM subset. (C and D) Frequency of cells coexpressing (C) HLA-DR, CD38, PD-1 and NKG2D or (D) NKG2D and granzyme B in EM compartment. (E) Mean percentage of NKG2D positive cells (left panel) or NKG2D/Granzyme B double positive cells (right panel) in HEV-3-, hCMV- and EBV-specific EM cells. (F and G) Plasmatic levels of indicated chemokines. Data represent mean values±S.E.M. $^{*}$P<0.05; $^{**}$P<0.01; $^{***}$P<0.001; $^{****}$P<0.0001. Symptomatic patients (S, red), asymptomatic patients (AS, black) and healthy controls (C, white).

**Table 2. Patient characteristics.**

|  | HEV-infected patients | Healthy controls |
|---|---|---|
|  | **n = 5** | **n = 4** |
| Gender (m/f) | 3/2 | 2/2 |
| Age (y) | 38 ± 3,6 | 39 ± 2,5 |
| Biochemical analysis |  |  |
| ALT Level (IU/L) | 1549 ± 368,8 |  |
| AST Level (IU/L) | 511,8 ± 229,1 |  |
| Bilirubin (μM) | 84,12 ± 40,5 |  |
| Anti-HEV IgM |  |  |
| Positive (n) | 5 | 0 |
| Negative (n) | 0 | 4 |
| Anti-HEV IgG |  |  |
| Positive (n) | 5 | 0 |
| Negative (n) | 0 | 4 |
| HEV RNA in plasma | 5.4 ± 0.2 |  |
| (log copies/ml) |  |  |
| HEV-3 subtypes | 3/2/0/0 |  |
| (3f/3c/3h/UD) |  |  |

n, number of subject; m, male; f, female; y, years; ALT, alanine aminotransferase; AST, aspartate aminotransferase; UD, undetermined. Data represent mean values ± S.E.M.

cell activation during HEV-3 infection. Increased T-bet expression further supports the involvement of these cytokines in EM cells bystander activation [20,38,39]. However, the source of IL-15 and IL-18 remains unclear. Infiltrating monocytes and tissue resident Kupffer cells are probably the main producers of these cytokines [30,40,41]. In the case of acute hepatitis A, hepatocytes have also been suggested as the main source of IL-15 [22], but we could not detect it in HEV-3-infected hepatocyte cell line culture. Further studies are thus needed to identify the source of these inflammatory cytokines.

The exuberant expression of T-Bet in symptomatic patients may also contribute to HEV-3-associated pathogenesis by controlling many aspect of inflammation [20]. Indeed, we showed that T-Bet expression is associated with increased levels of CXCR3 and granzyme B thereby orchestrating the recruitment, into the liver, of highly cytotoxic peripheral EM cells. In addition T-Bet, through the induction of CD122, potentiate the effect of IL-15 on EM cells. Although T-Bet is pivotal for IFN-γ production, the drop in TNF-α was unexpected, and future investigations would clarify whether T-Bet inhibits TNF-α production by disabling specific nuclear factors as previously described for IL-2 [42].

Cytokine-mediated activation of EM cells is a double-edged sword. EM cell activation might be beneficial to early pathogen clearance through IFN-γ production. But, it can also contribute to tissue damage by direct cytolysis of infected but also uninfected target cells through an NKG2D-mediated mechanism [22,26,29,30]. Indeed, NKG2D ligands are expressed not only on infected/transformed cells but also on neighboring healthy cells upon inflammatory microenvironment further promoting tissue injury. Thus, the significant increase in the frequency of NKG2D positive cells expressing granzyme B in symptomatic patients strongly suggests the involvement of EM cells in HEV-3 pathogenesis through their innate-like cytotoxicity.

Surprisingly, the activation of the EM compartment in symptomatic patients was associated with key features of early T cell exhaustion. Upregulation of iRs has been widely described as

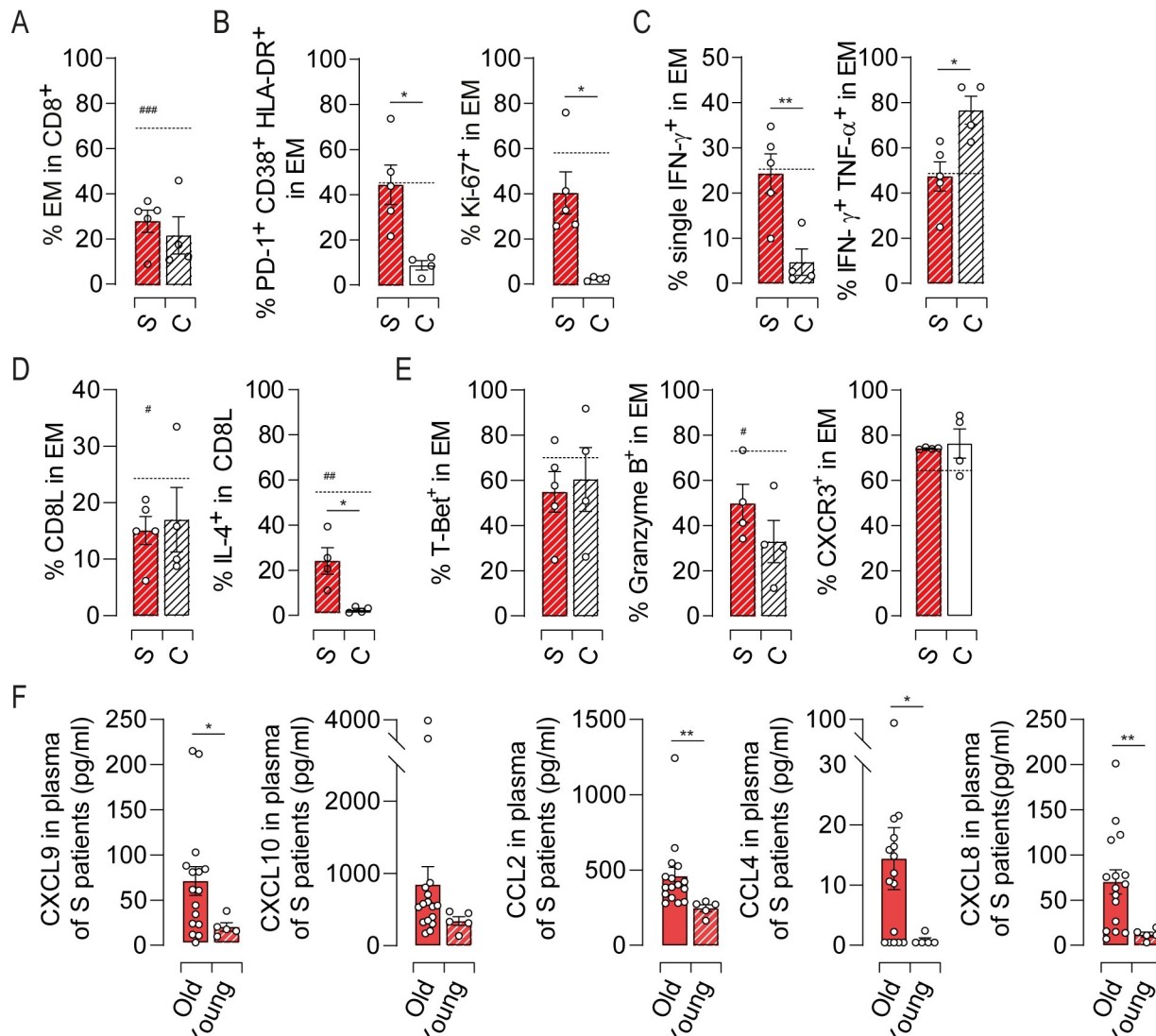

**Fig 8. Profile of the EM cell response in HEV-3 infected younger patients.** (A) Proportion of EM cells in CD8 T population. (B) Bar graphs representing the mean percentage of cells identified by triple expression of PD-1, HLA-DR and CD38 (left panel) and Ki-67 (right panel) in EM subset. (C) Bar graph illustrating the mean percentage of EM cells identified by single/double expression of IFN-γ and TNF-α. (D) Frequency of CD8L subset in EM compartment (left panel) and bar graphs representing the mean percentage of cells producing IL-4 in CD8L subset (right panel). (E) Expression profile of indicated markers in EM subset. (F) Plasmatic levels of indicated chemokines. Data represent mean values±S.E.M. Dotted line represent the mean of indicated markers in elderly symptomatic patients. # represent the statistical difference between younger and elderly patients. */#P<0.05; **P<0.01; ###P<0.001. Symptomatic elderly patients (S, red), younger patients (S, red hatch pattern) and young healthy controls (C, white hatch pattern).

the hallmark of T cell exhaustion in chronic infections such as chronic hepatitis B and C. Nevertheless, in healthy individuals, iRs provide negative feedback to limit overwhelming activation [15]. Moreover, strong upregulation of iRs was observed in experimental mouse models of both acute and persistent infections, but their sustained expression has been observed only in chronic settings [43]. Given that EM cell phenotype reach normal levels following patient recovery, the strong upregulation of iRs emphasizes here a robust ongoing CD8 T cell activation. On the other hand, EM cells displayed a partial loss in TNF-α production and a shift towards IFN-γ monofunctional cells. This issue may involve iRs and is not a correlate of

immune protection [44–47]. Since CD8 T cell cytokine network is not only crucial for viral clearance but also to orchestrating the response of other immune cells [32], the loss of cytokine polyfunctionality might contribute to HEV-3 pathogenesis.

Another aspect of symptomatic patients is the emergence of a CD8L subset that is able to produce IL-4 and less IFN-γ. Similar increase in the frequency of CD8L subset was previously described during the chronic phase of Hepatitis B virus infection [48]. This issue is in line with the exhaustion-like profile of the EM cell compartment in symptomatic patients. In addition to aging, persistent antigen presentation together with changes in the cellular metabolism and cytokine microenvironment can promote the differentiation of CD8L cells [10,48–52]. Although their exact role is unclear, IL-4-producing cells may serve as a negative feedback loop to dampen the intensity of the ongoing immune response. Indeed, IL-4 can decrease type-1 cytokines and inhibit the cytolytic function of neighboring CD8 T cells [53,54]. In agreement with this notion, IL-4 production negatively correlated with TNF-α and reverted to normal levels following the decline of the immune activation in symptomatic patients.

Although HEV-specific T-cell responses are detectable against non-structural proteins encoded in ORF1, most responses are detected against regions of the genome that are relatively well conserved, with the majority located in ORF2 [23–25,55]. Therefore, most of the experimental vaccines against hepatitis E are based on this antigen [56–58]. In agreement with these studies, we identified a weak frequency of HEV-3-specific cells in tested patients following stimulation with ORF3 peptides. Targeting ORF2-specific cells alone may underestimate the frequency of HEV-3-specific cells. However, even if we take into account the percentage of cells specific to ORF1 and ORF3, the global frequency of HEV-3-specific cells remains low and does not explain the drastic changes of the EM compartment in symptomatic patients. Thus, our initial hypothesis that bystander activation contributes to the observed alteration within the EM compartment remains valid.

In our study, HEV-3-infected males were most likely to develop clinical symptoms, with a male-to-female ratio of 2.2. Although consistent with other studies [1], this sex-dependent outcome remains unexplained since symptomatic patients shared similar phenotypic and functional alterations within the EM compartment regardless of their gender. Similarly, the asymptomatic man displayed the same profile than asymptomatic women. Besides genetics background, this discrepancy can be related to sex-hormones including androgens which can alter type-1/type-2 immune responses [59].

Another characteristic of HEV-3 is that older adults have greater risk for serious complications than young adults [1–3,60]. One possible explanation may be related to subclinical liver injury [60]. In agreement with other studies [11,49,61,62], our findings may also explain this discrepancy, since EM cells, which are more sensitive to bystander activation than other CD8 T cells subsets, were underrepresented and less shaped in young patients. Indeed, we showed here that aging is associated with increased inflammation, shift toward Th2 profile and high cytotoxic potential. These alterations participate to the decline of the immune system, resulting in increased vulnerability to infectious diseases.

In summary, we provide here in-depth characterization of the CD8 T cell response to HEV-3 infection in the elderly, both at the acute and convalescence phase. We also highlight metrics required for the development of efficient antiviral therapy including the nature/diversity of functional response. Although the sample size might be limiting due to difficulties in accessing asymptomatic patients, the sharp difference between study groups and the conserved phenotypic/functional profiles within each group outweighs this issue. To our knowledge, this is the first study to delineate immune correlates that contribute to HEV-3 pathogenesis and show that the global skewing of the EM compartment, regardless of antigen specificity, is associated with symptomatic infection in the elderly.

## Materials and methods

### Ethics statement

All patients provided written consent. The study was approved by the French South-West & Overseas ethical committee and was registered at the Ministry of Higher Education and Research (DC-2016-2772). Experiments were performed in agreement with the guidelines of the Declaration of Helsinki.

### Study population

Elderly and younger patients during the acute phase of HEV-3 infection were enrolled at the Toulouse University Hospital. Symptomatic patients developed liver-specific and nonspecific symptoms along with altered biochemical profile (Tables 1 and 2). Asymptomatic patients benefited from regular health monitoring. Following abnormal liver function tests including increased alanine aminotransferase levels (ALT), these patients were tested for HEV-3 infection and blood samples were taken straight after. There was no difference in the time of blood sampling after ALT peak between patient groups. Of note that the normal levels of ALT and AST are below 45 UI/L for men and 35 for women, and bilirubin is below 17μM. For three elderly symptomatic patients, blood samples were also collected one month after the onset of symptoms. Healthy donor samples were provided by the French National Blood Service. Elderly patients were all IgM negative and IgG positive for EBV. Except three subjects, they were also positive for hCMV. All samples were negative for HBV, HCV and HIV-1. None of the subjects had any history of liver disease or severe illness.

### HEV viremia and genotyping

Anti-HEV antibodies were detected using Wantai HEV-IgG and -IgM assays according to the manufacturer's instructions. RNA was extracted from blood samples using Total Nucleic Acid Isolation kit on the Cobas Ampliprep instrument (Roche Diagnostics). A real-time PCR technique amplifying a fragment in the ORF3 region was used for detection and quantification of viral RNA [63]. HEV-3 subtypes was determined by sequencing the ORF2 region [63].

### Peripheral blood mononuclear cells (PBMC) and plasma preparation

PBMC and plasma were obtained from blood samples by density gradient centrifugation using Ficoll-Paque Plus (Sigma-Aldrich). Cells were cryopreserved and plasma were stocked at -80˚C for further analysis.

### PBMC stimulation

PBMC ($10^6$ cells) were incubated for 3 hours or overnight with/without 50ng/ml phorbol-12-myristate-13-acetate and 1μM ionomycin (Sigma-Aldrich). To identify HEV-3-specific EM cells, PBMC were incubated overnight with 4μg/ml with a pool of peptides (15mers with 11aa overlap) that spans ORF2 or ORF3 (JPT). ORF2 and ORF3 pools are composed of 163 and 26 peptides respectively. DMEM:F-12 (V:V) medium supplemented with 10% fetal calf serum (FCS) was used for cell culture. To enable intracellular cytokine accumulation, BD GolgiPlug (BD Bioscience) was added in all conditions.

### Cell staining and multi-parameter flow cytometry analyses

Cells were washed with PBS and stained with Live/Dead Fixable Yellow Dead Cell Stain Kit for 30 minutes at room temperature (ThermoFisher Scientific). Nonspecific binding was blocked

by PBS supplemented with 10% human serum (HS). Surface staining was carried out in PBS supplemented with 1% HS for 30 minutes at 4˚C. For intracellular staining, cells were fixed and permeabilized using fixation/permeabilization kit (Ebioscience) according to the manufacturer's instructions. Intracellular staining was performed for 30 minutes at 4˚C. Cell phenotype was characterized by multi-color flow cytometry panels on a LSR-FORTESSA cytometer (BD Bioscience). Data were further analyzed using FlowJo v10 software (FlowJo LLC). Fluorescence-minus-one controls were used to determine positive staining. Geometric mean fluorescent intensity was calculated relatively to the negative staining for the concerned marker (ΔMFI).

## List of antibodies

For surface staining, the following antibodies were used: anti-CD3-PE-Vio615 (clone REA613), anti-CD8-VioBlue (clone BW135/80), anti-CCR7-PerCP-Vio700 (clone REA546), anti-CD45RA-VioGreen (clone REA562), anti-HLA-DR-APC-Vio770 (clone AC122), anti-CD38-PE-Vio770 (Clone REA572), anti-CD127-PE (clone MB15-18C9), anti-CD161-PE-Vio770 (clone 191B8), anti-TIM-3-FITC (clone F38-2E2), anti-LAG-3-APC (clone REA351), anti-CD27-VioBrightFITC (clone M-T271), anti-ICOS-APC-Vio770 (clone REA192), anti-CXCR3-APC (clone REA232), anti-CD122-APC (clone REA167), anti-NKG2D-PE or -Vio Bright-FITC (clone BAT221), anti-NKG2C-VioBrightFITC (clone REA205), anti-NKp44-Pe (clone 2.29) from Miltenyi biotec, anti-NKp30-AlexaFluor647 (clone P30-15), anti-NKp46-AlexaFluor647 (clone 9E2) from BioLegend and anti-PD-1-BV786 (clone EH12) from BD Bioscience. For HEV-3-unrelated CD8 T cell identification, PE-labeled MHC-I (HLA-A*02:01)-pentamer refolded with a CMV-pp65[495-504] or EBV- BMLF-1[259−267] epitopes were used according to the manufacturer's instructions (Proimmune). Intracellular staining was performed with the following antibodies: anti-IFN-γ-FITC (clone 4S.B3), anti-TNF-α-APC (clone Mab11), anti-IL-4-APC or -Pe-Cy7 (clone 8D4-8), anti-Bcl-2-FITC (clone Bcl2/100) from BD Bioscience, anti-Ki-67-FITC (clone REA183), anti-IL-2-PE (clone N7.48 A), anti-GATA-3-APC (clone REA174), granzyme B-FITC (clone REA226) from Miltenyi biotec, granzyme B-Pe-Cy7 (clone QA16A02) from BioLegend and anti-T-Bet-PE (clone 4B10) and anti-Eomes-APC-efluor780 (clone WD1928) from Thermofisher scientific.

## t-SNE analyses

t-SNE analyses were performed on three representative donors for each study group using FlowJo v10 software. For each donor, 5000 EM cells were selected using DownSample plugin. Cells were then merged using concatenate tool and barcoded for tracking. Finally, t-SNE analyses were performed using different markers (see figure legends). The heatmap represents the median intensity values for a given marker. A four-color scale was used with blue, green-yellow and red indicating low, intermediate and high expression levels respectively.

## Quantification of inflammatory cytokines and chemokines

Cytokines and chemokines were quantified by the LEGENDplex multi-Analyte Flow Assay Kit from BioLegend. IL-15 was assessed by the Quantikine ELISA test from R&D Systems according to the manufacturer's instructions.

## *In Vitro* HEV-3 infection

HepG2 human hepatoma cells (transfected stably with a MAVS-specific shRNA [64]) were infected for 10 days with an HEV-3 clinical strain obtained from the feces of an autochthone

infected patient at the acute phase of infection. Cells were then washed and cultured in 24-wells. Two days later, supernatant was harvested and considered for soluble factors quantification.

## Statistical analyses

Graphs represent mean values with error bars indicating the standard error of mean (S.E.M.). Statistical analyses and graphical representations were performed using GraphPad Prism v8.4.3 software. One-way analysis of variance (ANOVA) with the Newman-Keuls post-hoc test was used to compare subject groups. Two-tailed paired and unpaired *t*-tests were used to compare two parameters from the same or different donor, respectively. Correlations were evaluated by linear regression analyses and Spearman correlation test. Principal Component Analysis was performed using ClustVis (http://biit.cs.ut.ee/clustvis/). Values were centered, unit variance scaling was applied to rows and single value decomposition with imputation was used to calculate principal components. P values<0.05 were considered significant (*P<0.05, **P<0.01, ***P<0.001,****P<0.0001).

## Supporting information

**S1 Fig. Correlation between phenotypic changes and activation status in EM CD8 T cells.** (A) Percentage of cells expressing HLA-DR, CD38, PD-1, TIM-3, LAG-3, CD27, ICOS, CD127, CD161, Ki-67 and Bcl-2 in CD8 T subsets: Naïve (N), Effector (E) and Central Memory (CM). (B) Geometric Mean Fluorescence Intensity of indicated markers in effector memory (EM) CD8 T cells calculated relatively to the negative staining for the concerned marker (ΔMFI). (C) Correlation between the percentage of cells expressing HLA-DR or CD38 and CD127, CD161, PD-1, CD27 or ICOS by Spearman correlation test. (D) Percentage of EM CD8 T cells coexpressing PD-1 with TIM-3, and/or LAG-3. (E) Percentage of cells expressing Ki-67 and Bcl-2 based on the coexpression (+) or absence (-) of HLA-DR/CD38 within EM CD8 T cells from symptomatic patients. The Spearman correlation test P value and the R coefficient are indicated in each graph. Data represent mean values±S.E.M. *P<0.05; **P<0.01; ***P<0.001; ****P<0.0001. Symptomatic patients (S, red), asymptomatic patients (AS, black) and healthy controls (C, white). (TIF)

**S2 Fig. Association between type-1 cytokine production and activation status of EM CD8 T cells at different time of the infection.** (A) Representative dot plots for TNF-α production in EM CD8 T cells. (B) Mean percentage of EM CD8 T cells producing TNF-α after overnight polyclonal stimulation (PMA/Ionomycin). (C and D) Bar graphs illustrating the frequency of EM CD8 T cells producing (C) IFN-γ or (D) IL-2. (E) Percentage of cells expressing HLA-DR, CD38 or PD-1 in unstimulated (US) or PMA/Ionomycin (P/I) stimulated EM CD8 T subset from all study group. (F) Donut charts representing the mean percentage of cells expressing different combination of IFN-γ and TNF-α in EM CD8 T cells identified by the triple expression (Triple positive) of PD-1, HLA-DR and CD38. (G) Bar graphs representing statistical analyses of donut chart from Triple positive cells. (H) Donut charts representing the mean percentage of cells expressing different combination of IFN-γ and TNF-α in EM CD8 T cells identified by the single expression of CD38, HLA-DR and PD-1 or the absence of these markers (Triple negative). (I) Comparison of indicated parameters in three symptomatic patients at the onset of symptoms (Day 0, D0) and at the convalescence period, one month later (M1). Data represent mean values±S.E.M. *P<0.05; **P<0.01; ***P<0.001; ****P<0.0001. Symptomatic

patients (S, red), asymptomatic patients (AS, black) and healthy controls (C, white).
(TIF)

**S3 Fig. Phenotypic and functional analysis of EM CD8L subset.** (A) Representative histograms of IL-4 production in CD8 EM cells expressing high (CD8H) or low (CD8L) levels of CD8. (B) Mean percentage of IL-4-producing cells in the CD8H subset. (C) Percentage of IL-4-producing cells in unstimulated (US) or PMA/Ionomycin (P/I) stimulated CD8L subset from symptomatic patients. (D) Representative dot plots of IFN-γ and IL-4 expression in CD8L cells. (E) Correlation between TNF-α and IL-4 productions in EM CD8L T cells by Spearman correlation test. The Spearman correlation test P value and R coefficient are indicated in the graph. (F) Comparison of indicated parameters in three symptomatic patients at the onset of symptoms (Day 0, D0) and at the convalescence period, one month later (M1). Data represent mean values ± S.E.M. Symptomatic patients (S, red), asymptomatic patients (AS, black) and healthy controls (C, white).
(TIF)

**S4 Fig. Impact of T-bet expression on the function and phenotype of EM CD8 T cells.** (A) Mean percentage of EM CD8 T cells expressing Eomes (left panel) and Gata-3 (right panel). (B) Representative dot plots of T-Bet expression in the EM compartment. (C) Mean percentage of T-Bet positive cells within the EM CD8 T cells expressing high (CD8H) or low (CD8L) levels of CD8 from all study group. (D) Representative dot plots of T-Bet and IL-4 expression in EM cells. (E) Comparison of T-bet expression in three symptomatic patients at the onset of symptoms (Day 0, D0) and at the convalescence period, one month later (M1). (F and G) Frequency of cells identified by triple expression (left panel) or absence (right panel) of PD-1, HLA-DR and CD38 in the T-Bet positive and negative subset from AS patients (F) and Controls (G). (H and I) Bar graphs representing the mean percentage of cells from AS patients (H) and Controls (I) identified by single/double expression patterns of IFN-γ and TNF-α. Bar graphs represent statistical analyses of the donut charts (right panel). Data represent mean values±S.E.M. *P<0.05; **P<0.01; ***P<0.001;****P<0.0001. Symptomatic patients (S, red), asymptomatic patients (AS, black) and healthy controls (C, white).
(TIF)

**S5 Fig. Activation status of HEV3-specific and -unrelated EM CD8 T cells during HEV-3 infection.** (A) Frequency of IFN-γ positive cells in EM following stimulation with ORF2 or ORF3. (B and C) Representative dot plots of HLA-DR/CD38 coexpression (left panel) and PD-1 expression (right panel) in HEV-3 specific EM CD8 T cells from symptomatic (B) and asymptomatic (C) patients upon stimulation with ORF2 peptides. (D and E) Representative dot plots of HLA-DR/CD38 (left panel) and PD-1 (right panel) expression in hCMV specific EM CD8 T cells from symptomatic patients (D) and Controls (E). (F and G) Representative dot plots of HLA-DR/CD38 (left panel) and PD-1 (right panel) expression in EBV specific EM CD8 T cells from symptomatic patients (F) and Controls (G). Symptomatic patients (S, red).
(TIF)

**S6 Fig. Impact of inflammatory response to HEV-3 on the innate-like function of EM CD8 T cells.** (A) Frequency of cells expressing the indicated activating Natural Killer receptors in the EM compartment. (B) Frequency of granzyme B positive cells in HEV-3-, hCMV- and EBV-specific EM CD8 T cells. (C) Chemokine levels in HepG2 cell culture supernatant (I, infected; UI, uninfected). Symptomatic patients (S, red), asymptomatic patients (AS, black) and healthy controls (C, white).
(TIF)

**S7 Fig. Phenotypic landscape of CD8 T cell subset cells during HEV-3 infection in younger population.** (A) Mean percentage of CD8 T subsets: naïve (N), effector (E), effector memory (EM) and central memory (CM). (B) Percentage of cells expressing indicated markers in CD8 T subsets. Data represent mean values±S.E.M. *P<0.05; **P<0.01; ***P<0.001; ****P<0.0001. Younger patients (S, red hatch pattern) and young healthy controls (C, white hatch pattern). (TIF)

## Acknowledgments

The authors would like to thank the flow cytometry/immunomonitoring core facilities (CPTP-INSERM U1043) and D. McCarthy (English@Work) for English language proofreading.

## Author Contributions

**Conceptualization:** Hicham El Costa, Jacques Izopet.

**Formal analysis:** Hicham El Costa, Jordi Gouilly, Nabila Jabrane-Ferrat.

**Funding acquisition:** Hicham El Costa.

**Investigation:** Hicham El Costa, Jordi Gouilly.

**Methodology:** Hicham El Costa.

**Resources:** Florence Abravanel, Jean-Marie Peron, Nassim Kamar.

**Writing – original draft:** Hicham El Costa.

**Writing – review & editing:** Elmostafa Bahraoui, Nabila Jabrane-Ferrat, Jacques Izopet.

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
