## [Decision Letter · Decision Letter 0]

20 Aug 2020

Dear Dr. EL COSTA,

Thank you very much for submitting your manuscript "Effector memory CD8 T cell response elicits Hepatitis E Virus genotype 3 pathogenesis in the elderly" for consideration at PLOS Pathogens. As with all papers reviewed by the journal, your manuscript was reviewed by members of the editorial board and by several independent reviewers. In light of the reviews (below this email), we would like to invite the resubmission of a significantly-revised version that takes into account the reviewers' comments.

Two major weaknesses were identified by the reviewers. First, the conclusion that bystander CD8+ T cell activity causes more severe disease in elderly subjects is not well supported in the absence of comparative data from younger symptomatic subjects. Second, the study does not provide insight into the mechanism of CD8+ T cell pathogenesis in these subjects, which is required to understand and support the case for their role in the pathogenesis of liver injury. The reviewers questions must be addressed by adding relevant experimental data to a revised manuscript.

We cannot make any decision about publication until we have seen the revised manuscript and your response to the reviewers' comments. Your revised manuscript is also likely to be sent to reviewers for further evaluation.

Sincerely,

Christopher M. Walker

Associate Editor

PLOS Pathogens

Jing-hsiung James Ou

Section Editor

PLOS Pathogens

Kasturi Haldar

Editor-in-Chief

PLOS Pathogens

orcid.org/0000-0001-5065-158X

Michael Malim

Editor-in-Chief

PLOS Pathogens

orcid.org/0000-0002-7699-2064

Two major weaknesses were identified by the reviewers. First, the conclusion that bystander CD8+ T cell activity causes more severe disease in elderly subjects is not well supported in the absence of comparative data from younger symptomatic subjects. Second, the study does not provide insight into the mechanism of CD8+ T cell pathogenesis in these subjects, which is required to understand and support the case for their role in the pathogenesis of liver injury. The reviewers questions must be addressed by adding relevant experimental data to a revised manuscript.

Reviewer's Responses to Questions

**Part I - Summary**

Reviewer #1: This study provides valuable information regarding CD8 T cell responses in HEV genotype 3 infection. However, this study is descriptive with a lack of a mechanistic insight. Particularly, this study strongly indicate that HEV-3-unleated memory CD8 T cells undergo bystander activation in the absence of cognate antigens. This issue is very important because the expansion of effector memory CD8 T cells might be caused mainly by bystander activation, not by HEV-3 stimulation. It is better to perform more detailed analysis for this important issue as commented below.

Reviewer #2: In the presented manuscript, Costa et al. examine the role of CD8 T cells among twenty immunocompetent elderly subjects, including twelve HEV-3-infected patients. Using multi-parametric flow cytometry, the authors observe that that symptomatic patients display a strong activation of HEV-3-specific and -non-specific EM CD8 T cells associated with qualitative and quantitative alterations in cytokine production. In particular, the authors observe that symptomatic patients displayed a drastic drop of TNF-α suggesting an intrinsic defect in effector memory (EM) cells.

Overall, the content of the manuscript is well written and structured and seems suitable for the readership of the journal. However, while the analysis of immunopathogenesis of Hepatitis E virus infection in elderly patients is quite novel and the experiments are conducted according to high technical standards, the cohort of selected patients currently limits the overall relevance of the study.

Reviewer #3: El Costa and co-workers analyze the global CD8+ T cell phenotype in elderly patients with hepatitis E virus genotype 3 (HEV-3) infection by comparing 7 elderly patients with acute-symptomatic HEV-3, 5 elderly patients with asymptomatic HEV-3 and 8 elderly healthy controls. They find an association of symptomatic HEV-3 with a high frequency of activated (HLA-DR+ PD1+ CD38+) effector memory CD8+ T cells that are skewed towards mono-functionality (e.g. IFNy production without TNF production) and display high T-bet expression. In addition, they find an enhanced fraction of CD8+ T cells to have Tc2 commitment. Lastly, the authors argue that a low proportion of these activated effector memory T cells are HEV-3-specific, arguing for a high bystander effect in symptomatic elderly patients.

**Part II – Major Issues: Key Experiments Required for Acceptance**

Reviewer #1: 1. The authors need to examine further the phenotype and functional characteristics of bystander-activated CD8 T cells. For example, T-bet expression, loss of TNF production, and increased production of IL-4 need to be examined in the gate of bystander-activated CD8 T cells (e.g. CMV-HLA-A2 pentamer+CD8+ T cells). Regarding bystander activation of memory CD8 T cells, please show additional examples other than CMV-HLA-A2 pentamer+CD8+ T cells (e.g. EBV-specific or flu-specific).

2. In addition, the expression of NK receptors need to examined in HEV-3-specific CD8 T cells and bystander-activated CD8 T cells (e.g. CMV-HLA-A2 pentamer+CD8+ T cells), including NKG2D. It was previously shown that NKG2D is upregulated only in IL-15-induced bystander-activated CD8 T cells, not in TCR-stimulated CD8 T cells (Kim J et al. Immunity. 2018 Jan 16;48(1):161-173.e5. doi: 10.1016/j.immuni.2017.11.025.).

3. For functions of effector memory CD8 T cells, the authors focused on cytokine production and cytokine polyfunctionality. However, it is also required to investigate a cytotoxic capacity of effector memory CD8 T cells by staining perforin and granzymes.

Reviewer #2: The overall sample size of the studied cohort is very small (n=20), in particular, the symptomatic patients seem to be significantly older than the other two groups. The authors are therefore advised to include further subjects with similar age into their study. In particular, the samples of elderly asymptomatic patients and control patients.

Furthermore, since the authors do not include longitudinal samples, it remains unclear, whether the loss of cytokine production (i.e. TNFalpha) is just occurs as a temporal phenomenon or also remains after the patients clear the virus.

The authors only employ samples from older patients, however it seems relevant to compare the observed effects of HEV on the composition on immune cells to samples of younger patients to directly allow conclusions regarding age-related effects of an HEV infection.

Reviewer #3: In general, this is an interesting and important research topic and the authors are (to their and my knowledge) the first to address the often severe pathogenesis of HEV-3 in the elderly. There are, however, several major limitations that need to be at least carefully discussed in a revised version:

1. The authors had the aim to determine the mechanisms underlying HEV-3 pathogenesis in the elderly. Is the pathogenesis of symptomatic HEV-3 in the elderly different from that in younger patients? Why did the authors not include a control group of young patients with acute-symptomatic HEV-3?

2. Is the activation of effector memory cells in acute-symptomatic HEV-3 in the elderly different from other viral infections such as Flu? Absolute changes in global T cell phenotypes are different to interpret without comparison to better understood settings. In other words, is this just a normal reaction to an acute viral infection or is this unique to HEV-3?

3. HEV-3-specific CD8+ T cell were only estimated by using ORF2-derived overlapping peptides, so that ORF1- and ORF3-specific CD8+ T cell were not analyzed. This might lead to an important bias in the assessment of HEV3-specific versus bystander effects.

4. CMV-specific CD8+ T cells were activated in acute-symptomatic HEV3. The authors interpret this as bystander effect. This could also be a result of heterologous immunity (compare Soon et al., Frontiers in Immunology 2019).

**Part III – Minor Issues: Editorial and Data Presentation Modifications**

Reviewer #1: 1. For IL-4 ICS, it seems that PBMCs were cultured without ex vivo stimulation (US in Figure S4C). However, I cannot find a detailed method for this assay. It is important because ICS is usually performed with ex vivo stimulation.

2. In Figure 2E, I understood that ‘% HLA-DR(+)’ means ‘% HLA-DR(+)CD38(-)PD-1(-)’. If this is the case, please clearly re-write the y-axis. (same for ‘% CD38+’ and ‘% PD-1+’).

Reviewer #2: The authors frequently use s.e.m. during their boxplots and might consider to display values of the individual patients throughout the manuscript.

Its not clear how the cluster of patients were identified during the PCA analysis in Fig.1F.

Reviewer #3: Minor:

1. Table 1: 1/5 patients with “asymptomatic” course had fever. Did he have signs of a bacterial infection focus? Otherwise, he/she had a symptomatic course by definition?

2. Table 1: Please state normal ranges especially for bilirubin (since the micromolar range is not that common).

3. Figures, e.g. Fig. 2e: Please change “%HLA-DR+ in EM” to e.g. “% only HLA-DR+ in EM” to make clear that single-positive cells are meant here.

4. Fig. 3b: light/dark blue (IFNy+TNF+ and IL2+, respectively) and light/dark purple (TNF+IL2+ and IFNy+IL2+, respectively) are difficult/impossible to distinguish.

PLOS authors have the option to publish the peer review history of their article (what does this mean?). If published, this will include your full peer review and any attached files.

Reviewer #1: **Yes: **Eui-Cheol Shin

Reviewer #2: No

Reviewer #3: No
---

## [Editor Report · Decision Letter 1]

9 Feb 2021

Dear Dr. EL COSTA,

We are pleased to inform you that your manuscript 'Effector memory CD8 T cell response elicits Hepatitis E Virus genotype 3 pathogenesis in the elderly' has been provisionally accepted for publication in PLOS Pathogens.

Best regards,

Christopher M. Walker

Associate Editor

PLOS Pathogens

Jing-hsiung James Ou

Section Editor

PLOS Pathogens

Kasturi Haldar

Editor-in-Chief

PLOS Pathogens

orcid.org/0000-0001-5065-158X

Michael Malim

Editor-in-Chief

PLOS Pathogens

orcid.org/0000-0002-7699-2064
---

## [Editor Report · Acceptance letter]

18 Feb 2021

Dear Dr. El Costa,

We are delighted to inform you that your manuscript, "Effector memory CD8 T cell response elicits Hepatitis E Virus genotype 3 pathogenesis in the elderly," has been formally accepted for publication in PLOS Pathogens.

Best regards,

Kasturi Haldar

Editor-in-Chief

PLOS Pathogens

orcid.org/0000-0001-5065-158X

Michael Malim

Editor-in-Chief

PLOS Pathogens

orcid.org/0000-0002-7699-2064